# Towards Provable Control for Unknown Linear Dynamical Systems

## Abstract

We study the control of symmetric linear dynamical systems with unknown dynamics and a hidden state. Using a recent spectral filtering technique for concisely representing such systems in a linear basis, we formulate optimal control in this setting as a *convex* program. This approach eliminates the need to solve the non-convex problem of explicit identification of the system and its latent state, and allows for provable optimality guarantees for the control signal. We give the first efficient algorithm for finding the optimal control signal with an arbitrary time horizon $T$, with sample complexity (number of training rollouts) polynomial only in $\log T$ and other relevant parameters.

## 1 Introduction

Recent empirical successes of reinforcement learning involve using deep nets to represent the underlying MDP and policy. However, we lack any supporting theory, and are far from developing algorithms with provable guarantees for such settings. We can make progress by addressing simpler setups, such as those provided by control theory.

Control theory concerns the control of dynamical systems, a non-trivial task even if the system is fully specified and provable guarantees are not required. This is true even in the simplest setting of a linear dynamical system (LDS) with quadratic costs, since the resulting optimization problems are high-dimensional and sensitive to noise.

The task of controlling an unknown linear system is significantly more complex, often giving rise to non-convex and high-dimensional optimization problems. The standard practice in the literature is to first solve the non-convex problem of *system identification*—that is, recover a model that accurately describes the system—and then apply standard robust control methods. The non-convex problem of system identification is the main reason that we have essentially no provable algorithms for controlling even the simplest linear dynamical systems with unknown latent states.

In this paper, we take the first step towards a provably efficient control algorithm for linear dynamical systems. Despite the highly non-convex and high-dimensional formulation of the problem, we can efficiently find the optimal control signal in polynomial time with optimal sample complexity. Our method is based on *wave-filtering*, a recent spectral representation technique for symmetric LDSs (Hazan et al., 2017).

### 1.1 Our results

A *dynamical system* converts input signals $\{x_1, \ldots, x_T\} \in \mathbb{R}^n$ into output signals $\{y_1, \ldots, y_T\} \in \mathbb{R}^m$, incurring a sequence of costs $c_1, \ldots, c_T \in \mathbb{R}$. We are interested in controlling unknown dynamical systems with hidden states (which can be thought of as being partially "observed"via the output signals). A vast body of work focuses on linear dynamical systems with quadratic costs, in which the $\{y_t\}$ and $\{c_t\}$ are governed by the following dynamics:

$$h_{t+1} = Ah_t + Bx_t, \quad y_t = Ch_t + Dx_t + \eta_t, \quad c_t = y_t^\top Q y_t + x_t^\top R x_t,$$

where $h_1, \ldots, h_T \in \mathbb{R}^d$ is a sequence of hidden states starting with a fixed $h_1$. Matrices $A, B, C, D, Q, R$ of appropriate dimension describe the system and cost objective; the $\{\eta_t\}$ are Gaussian noise vectors. All of these matrices, as well as the parameters of the Gaussian, can be

unknown. The most fundamental control problem involves controlling the system for some time horizon $T$: find a signal $x_1, \ldots, x_T$ that minimizes the sum of these quadratic output costs $\sum_t c_t$. Clearly, any algorithm for doing so must first learn the system parameters in some form, and this is often the source of computational intractability (meaning algorithms that take time exponential in the number of system parameters).

Previously known algorithms are of two types. The first type tries to solve the non-convex problem, with algorithms that lack provable guarantees and may take exponential time in the worst case: e.g., expectation-maximization (EM) or gradient-based methods (back-propagation through time, like the training of RNNs) which identify both the hidden states and system parameters.

Algorithms of the second type rely upon regression, often used in time-series analysis. Since $T$-step dynamics of the system involve the first $T$ powers of $A$, the algorithm represents these powers as new variables and learns the $T$-step dynamics via regression (e.g., the so-called VARX$(p, s)$ model) assuming that the system is well-conditioned; see Appendix A. This has moderate computational complexity but high sample complexity since the number of parameters in the regression scales with $T$, the number of time steps.

Our new method obtains the best of both results: few parameters to train resulting in low sample complexity, as well as polynomial computational complexity. Table 1 below compares the different methods, where we denote by $|\mathfrak{D}| = \max(\|B\|_F, \|C\|_F, \|D\|_F, \|Q\|_{\mathrm{op}}, m, n)$ the size of the system, and by $T$ the time horizon for planning.

| Method | Sample complexity | Time complexity | Assumptions |
|---|---|---|---|
| System Identification | $\mathrm{poly}(\epsilon^{-1}, |\mathfrak{D}|)$ | exponential | none |
| VARX$(0, s)$ | $\mathrm{poly}(\epsilon^{-1}, |\mathfrak{D}|, s)$ | $\mathrm{poly}(T, |\mathfrak{D}|)$ | $s = \Omega\left(\frac{1}{1 - \|A\|_2}\right)$ or $s = T$ |
| Ours | $\mathrm{poly}(\epsilon^{-1}, |\mathfrak{D}|, \mathbf{\log T})$ | $\mathrm{poly}(T, |\mathfrak{D}|)$ | $A$ symmetric |

Table 1: Comparison of control algorithms.

In Section 2 we state the precise algorithm. The informal result is as follows.

---

**Theorem 1.1** (Controlling an unknown LDS; informal). *Let $\mathfrak{D}$ be a linear dynamical system with a symmetric transition matrix $A$ and with size $|\mathfrak{D}|$. Then, for every $\varepsilon > 0$, Algorithm 1 produces a sequence of controls $(\hat{x}_1, \ldots \hat{x}_T)$, $\|\hat{x}_i\|_2 \leq 1$ with $\|\hat{x}_{1:t}\|_2 \leq L$, such that*

$$\mathbb{E}\left[\frac{1}{T}\sum_{t=1}^{T} \mathrm{cost}_{\mathfrak{D}}^t(\hat{x}_{1:t})\right] \leq \min_{\substack{x_{1:T} \in \mathbb{B}_2^T \\ \|x_{1:T}\|_2 \leq L}} \mathbb{E}\left[\frac{1}{T}\sum_{t=1}^{T} \mathrm{cost}_{\mathfrak{D}}^t(x_{1:t})\right] + \varepsilon. \tag{1}$$

*Assuming i.i.d. Gaussian noise $\eta_t \sim \mathcal{N}(0, \Sigma)$, the algorithm samples $\tilde{O}\left(\mathrm{poly}\left(|\mathfrak{D}|, L, \mathrm{Tr}\,\Sigma, 1/\varepsilon\right)\right)$ trajectories from $\mathfrak{D}$, and runs in time polynomial in the same parameters.*

---

### 1.2 RELATED WORK

The field of optimal control for dynamical systems is extremely broad and brings together literature from machine learning, statistical time-series analysis, dynamical system tracking and Kalman filtering, system identification, and optimal control. For an extensive survey of the field, see e.g. (Todorov, 2006; Bertsekas, 2000).

**Tracking a known system.** A less ambitions goal than control is tracking of a dynamical system, or prediction of the output given a known input. For the special case of LDS, the well-known Kalman filter (Kalman, 1960) is an optimal recursive least-squares solution for maximum likelihood estimation (MLE) under Gaussian perturbations to a linear dynamical system.

**System identification.** When the underlying dynamical system is unknown, there are essentially no provably efficient methods for recovering it. For various techniques used in practice, see the

classic survey (Ljung, 1998). Roweis & Ghahramani (1999) suggest using the EM algorithm to learn the parameters of an LDS, nowadays widespread, but it is well-known that optimality is not guaranteed. The recent result of Hardt et al. (2016) gives a polynomial time algorithm for system recovery, although it applies only to the single-input-single-output case and makes various statistical assumptions on the inputs.

**Model-free tracking.** Our methods depend crucially on a new algorithm for LDS sequence prediction, at the heart of which is a new convex relaxation for the tracking formulation (Hazan et al., 2017). In particular, this method circumvent the obstacle of explicit system identification. We detail our usage of this result in Definition 2.3.

We note an intriguing connection to the recently widespread use of deep neural networks to represent an unknown MDP in reinforcement learning: the main algorithm queries the unknown dynamical system with *exploration* signals, and uses its responses to build a compact representation (denoted by $\hat{M}$ in Algorithm 1) which estimates the behavior of the system.

**Time-series analysis.** One of the most common approaches to modeling dynamical systems is the autoregressive-moving average (ARMA) model and its derivatives in the time-series analysis literature (Hamilton, 1994; Box et al., 1994; Brockwell & Davis, 2009). At the heart of this method is the autoregressive form of a time series, namely,

$$x_t = \sum_{i=1}^{k} A_i x_{t-i} + \varepsilon_t.$$

Using online learning techniques, it is possible to completely identify an autoregressive model, even in the presence of adversarial noise (Anava et al., 2013). This technique lies at the heart of a folklore regression method for optimal control, given in the second row of table 1.

**Optimal control.** The most relevant and fundamental primitive from control theory, as applied to the control of linear dynamical systems, is the *linear-quadratic-Gaussian* (LQG) problem. In this setting, the system dynamics are assumed to be known, and the task is to find a sequence of inputs which minimize a given quadratic cost. A common solution, the LQG controller, is to combine Kalman filtering with a *linear-quadratic regulator*, a controller selected by solving the Bellman equation for the problem. Such an approach admits theoretical guarantees under varied assumptions on the system; see, for example, Dean et al. (2017).

Our setting also involves a linear dynamical system with quadratic costs, and thus can be seen as a special case of the LQG setup, in which the process noise is zero, and the transition matrix is assumed to be symmetric. However, our results are not analogous: our task also includes learning the system's dynamics. As such, our main algorithm for control takes a very different approach than that of the standard LQR: rather than solving a recursive system of equations, we provide a formulation of control as a one-shot convex program.

## 2 STATEMENT OF MAIN THEOREM

First, we state the formal definitions of the key objects of interest.

**Definition 2.1** (Dynamical system). *A dynamical system $\mathfrak{D}$ is a mapping that takes a sequence of input vectors $x_1, \ldots, x_T \in \mathbb{B}_2 = \{x \in \mathbb{R}^n : \|x\|_2 \leq 1\}$ to a sequence of output vectors $y_1, \ldots, y_T \in \mathbb{R}^m$ and costs $c_1, \ldots, c_T \in \mathbb{R}$. Denote $x_{s:t} = [x_s ; \ldots ; x_t]$ as the concatenation of all input vectors from time $s$ to $t$, and write*

$$\mathfrak{D}^t(x_{1:t}) = y_t \ , \ \mathsf{cost}_{\mathfrak{D}}^t(x_{1:t}) = c_t.$$

**Definition 2.2** (Linear dynamical system). *A linear dynamical system (LDS) is a dynamical system whose outputs and costs are defined by*

$$
\begin{aligned}
h_{t+1} &= A h_t + B x_t, \\
y_t &= C h_t + D x_t + \eta_t && \text{with } \eta_t \sim \mathcal{N}(0, \Sigma), \\
c_t &= y_t^\top Q y_t + x_t^\top R x_t,
\end{aligned}
$$

*where $h_1, \ldots, h_T \in \mathbb{R}^d$ is a sequence of hidden states starting with fixed $h_1$, and $A, B, C, D, Q, R$ are matrices (or vectors) of appropriate dimension. We assume $\|A\|_{op} \leq 1$, i.e., all singular values of $A$ are at most one, and that $Q \succcurlyeq 0, R \succcurlyeq 0$.*

Our algorithm and its guarantees depend on the construction of a family of orthonormal vectors in $\mathbb{R}^T$, which are interpreted as convolution filters on the input time series. We define the *wave-filtering matrix* below; for more details, see Section 3 of Hazan et al. (2017).

**Definition 2.3** (Wave-filtering matrix). *Fix any $n, T$, and $1 \leq k \leq T$. Let $\phi_j$ be the eigenvector corresponding to the $j$-th largest eigenvalue of the Hankel matrix $Z_T \in \mathbb{R}^{T \times T}$, with entries $Z_{ij} := \frac{2}{(i+j)^3 - (i+j)}$. The wave-filtering matrix $\Phi \in \mathbb{R}^{nk \times nT}$ is defined by $k$ vertically stacked block matrices $\{\Phi^{(j)} \in \mathbb{R}^{n \times nT}\}$, defined by horizontally stacked multiples of the identity matrix:*

$$\Phi^{(j)} \stackrel{def}{=} \left[ \phi_j(1) \, I_n \quad \phi_j(2) \, I_n \quad \ldots \quad \phi_j(T) \, I_n \right].$$

Then, letting $t$ range from 1 to $T$, $\Phi x_{t:t-T}$ then gives a dimension-wise convolution of the input time series by the filters $\{\phi_j\}$ of length $T$. Theorem 3.3 uses a structural result from Hazan et al. (2017), which guarantees the existence of a concise representation of $\mathfrak{D}$ in the basis of these filters.

The main theorem we prove is the following.

---

**Theorem 2.4** (Controlling an unknown LDS). *Let $\mathfrak{D}$ be a LDS with a symmetric transition matrix $A$ and with $\|B\|_F, \|C\|_F, \|D\|_F, \|Q\|_{op} \leq \rho$, and with $Q \succcurlyeq \lambda I$. For every $\varepsilon > 0$, Algorithm 1, with a choice of $k = \Omega\left( \log^2 T \log\left( \frac{n\rho k}{\lambda \varepsilon} \right) \right)$ and $\Omega\left( \frac{L^2 \rho^2 nkc \operatorname{Tr}(\Sigma) \ln\left( \frac{T}{\delta} \right)}{\lambda \varepsilon^2} \right)$, produces a sequence of controls $(\hat{x}_1, \ldots \hat{x}_T) \in \mathbb{B}_2^T$, such that with probability at least $1 - \delta$,*

$$\mathbb{E}\left[ \frac{1}{T} \sum_{t=1}^{T} \operatorname{cost}_{\mathfrak{D}}^t(\hat{x}_{1:t}) \right] \leq \min_{\substack{x_{1:T} \in \mathbb{B}_2^T \\ \|x_{1:T}\| \leq L}} \mathbb{E}\left[ \frac{1}{T} \sum_{t=1}^{T} \operatorname{cost}_{\mathfrak{D}}^t(x_{1:t}) \right] + \varepsilon, \tag{2}$$

*assuming that*

$$\min_{\substack{x_{1:T} \in \mathbb{B}_2^T \\ \|x_{1:T}\| \leq L}} \mathbb{E}\left[ \frac{1}{T} \sum_{t=1}^{T} \operatorname{cost}_{\mathfrak{D}}^t(x_{1:t}) \right] \leq \operatorname{Tr}(Q\Sigma) + c \tag{3}$$

*Further, the algorithm samples* $\operatorname{poly}\left( \frac{1}{\varepsilon}, \log\left( \frac{1}{\delta} \right), \log T, \log \rho, \frac{1}{\lambda}, L, n, k, \operatorname{Tr}(\Sigma) \right)$ *trajectories from the dynamical system, and runs in time polynomial in the same parameters.*

---

We remark on some of the conditions. $\lambda$ is bounded away from 0 when we suffer loss in all directions of $y_t$. In condition (3), $\operatorname{Tr}(Q\Sigma)$ is inevitable loss due to background noise, so (3) is an assumption on the system's controllability.

We set up some notation for the algorithm. Let $\Phi \in \mathbb{R}^{nk \times nT}$ be the *wave-filtering* matrix from Definition 2.3. Let $x_i = 0$ for $i \leq 0$, and let $X_t = x_{t:t-T+1}$. Let $\rho = \max(\|B\|_F, \|C\|_F, \|D\|_F, \|Q\|_F)$ be an upper bound for the matrices involved in generating the outputs and costs.

## 3 PROOF OF MAIN THEOREM

To prove Theorem 2.4, we invoke Lemma 3.1 and Lemma 3.2, proved in Subsection 3.1 and Subsection 3.2, respectively.

**Lemma 3.1** (Learning dynamics). *For every $\varepsilon > 0$ and symmetric LDS $\mathfrak{D}$, with probability $\geq 1 - \delta$, setting $k = \Omega(\log^2 T \log(n\rho/\varepsilon))$ and $S = \Omega\left( \frac{L^2 nk \operatorname{Tr}(\Sigma) \ln\left( \frac{T}{\delta} \right)}{\varepsilon^2} \right)$, step 7 in Algorithm 1 computes a matrix $\hat{M} \in \mathbb{R}^{m \times nk}$ such that for every sequence of input signals $(x_1, \ldots, x_T) \in \mathbb{B}_2^T$ satisfying $\|x_{1:T}\|_2 \leq L$ and $1 \leq t \leq T$, we have that*

$$\|\hat{y}_t - \mathbb{E} y_t\| \leq \varepsilon \tag{5}$$

$$\text{where } \hat{y}_t = \hat{M} \Phi X_t + z_t \text{ and } y_t = \mathfrak{D}_y^t(x_{1:t}). \tag{6}$$

---

**Algorithm 1:** Control with an LDS oracle

---

**Input** : Oracle access to LDS $\mathfrak{D}$, cost matrices $Q, R$, filter parameter $k$, sample count $S$.
**Output:** Control inputs $\hat{x}_{1:T}$.

**1** Run $\mathfrak{D}$ with the all-zeros input $S$ times, recording responses $z_{1:T}^{(s)} := \mathfrak{D}^{1:T}(0)$ for each $1 \leq s \leq S$.

**2** Average the zero-impulse responses: $z_{1:T} := \frac{1}{S} \sum_{s=1}^{S} z_{1:T}^{(s)}$.

**3 for** $1 \leq j \leq nk$ **do**

**4** $\quad$ Run $\mathfrak{D}$ with input $\phi_j$ $S$ times, recording responses $y_T^{(s,j)} := \mathfrak{D}^T(\phi_j)$ for each $1 \leq s \leq S$.

**5** $\quad$ Average the exploration responses: $y_T^{(j)} := \frac{1}{S} \sum_{s=1}^{S} y_T^{(s,j)}$.

**6 end**

**7** Let $\hat{M}$ be the matrix whose $j$-th column is $\hat{M}_j := y_T^{(j)} - z_T$.

**8** Solve the following convex program to obtain controls $(\hat{x}_1, \ldots \hat{x}_T)$:

$$\hat{x}_{1:T} := \underset{\substack{x_{1:T} \in \mathbb{B}_2^T \\ \|x_{1:T}\|_2 \leq L}}{\arg\min} \sum_{t=1}^{T} \left[ (\hat{M}\Phi X_t + z_t)^\top Q(\hat{M}\Phi X_t + z_t) + x_t^T R x_t \right]. \tag{4}$$

---

**Lemma 3.2** (Robustness of control to uncertainty in dynamics). *Let*

$$\hat{x}_{1:T} = \underset{\substack{x_{1:T} \in \mathbb{B}_2^T \\ \|x_{1:T}\| \leq L}}{\arg\min} \sum_{t=1}^{T} \left[ \hat{\mathfrak{D}}^t(x_{1:t})^\top Q \hat{\mathfrak{D}}^t(x_{1:t}) + x_t^T R x_t \right],$$

*where $\hat{M} \in \mathbb{R}^{m \times nk}$ is such that for every sequence of input signals $(x_1, \ldots, x_T) \in \mathbb{B}_2^T$ with $\|x_{1:T}\|_2 \leq L$ and $t \leq T$, Equation (5) holds with $\hat{y}_t = \hat{\mathfrak{D}}^t(x_{1:t})$. Assume (3). Then*

$$\mathbb{E}\left[ \frac{1}{T} \sum_{t=1}^{T} \mathsf{cost}_{\mathfrak{D}}^t(\hat{x}_{1:t}) \right] \leq \min_{\substack{x_{1:T} \in \mathbb{B}_2^T \\ \|x_{1:T}\| \leq L}} \mathbb{E}\left[ \frac{1}{T} \sum_{t=1}^{T} \mathsf{cost}_{\mathfrak{D}}^t(x_{1:t}) \right] + O\left( \sqrt{\frac{c}{\lambda}} \rho \varepsilon \right), \tag{7}$$

*Moreover, the minimization problem posed above is convex (for $Q, R \succcurlyeq 0$), and can be solved to within $\varepsilon_{\mathrm{opt}}$ accuracy in $\mathrm{poly}(T, \rho, 1/\varepsilon_{\mathrm{opt}})$ time using the ellipsoid method.*

*Proof of Theorem 2.4.* Use Lemma 3.1 with $\varepsilon \hookleftarrow O\left( \sqrt{\frac{\lambda}{c}} \frac{\varepsilon}{\rho} \right)$. Note that $x_{1:T} = \phi_j$ is a valid input to the LDS because $\|\phi_j\|_2 = 1$. Now use Lemma 3.2 on the conclusion of Lemma 3.1. $\qquad \square$

### 3.1 LEARNING THE DYNAMICS

To prove Lemma 3.1, we will use the following structural result from Hazan et al. (2017) restated to match the setting in consideration.

**Theorem 3.3.** *Let $\mathfrak{D}$ be a symmetric LDS with the stated conditions from Section 2, with fixed $h_1$. Let $\varepsilon > 0$, and let $k = \Omega(\log^2 T \log(n\rho/\varepsilon))$. Let $z_{1:t}$ be the output if the input signal is 0. Let $M'$ be the matrix such that for all input-output pairs $(x_{1:T}, y_{1:T})$, $\mathbb{E}(y_T - z_T) = M' X_T$. Then for all $x_{1:T} \in \mathbb{B}_2^\top$, the matrix $M = M' \Phi^\top$ satisfies*

$$\|\mathbb{E}(y_t - z_t) - M\Phi X_t\| \leq \varepsilon, \qquad\qquad \forall t \leq T. \tag{8}$$

*Proof.* This follows from Theorem 3b in Hazan et al. (2017) after noting four things.

1. $\mathbb{E}(y_t - z_t)$ are the outputs when the system is started at $h_1 = 0$ with inputs $x_{1:t}$ and no noise. A linear relationship $y_t - z_t = M' X_t$ holds by unfolding the recurrence relation.

2. Examining the construction of $M$ in the proof of Theorem 3b, the $M$ is exactly the projection of $M'$ onto the subspace defined by $\Phi$. (Note we are using the fact that the rows of $\Phi$ are orthogonal.)

3. Theorem 3b is stated in terms of the quantity $L_y$, which is bounded by $\rho^2$ by Lemma F.5 in Hazan et al. (2017).

4. We can modify the system so that $D = O$ by replacing $(A, B, C, D)$ with $\left(\left(\begin{smallmatrix} A & O \\ O & O \end{smallmatrix}\right), \left(\begin{smallmatrix} B \\ I \end{smallmatrix}\right), (C\,D), O\right)$. This replaces $\rho$ by $O(\max(\rho, \sqrt{n}))$. Theorem 3b originally had dependence of $y_t$ on both $\Phi X_t$ and $x_t$, but if $D = O$ then the dependence is only on $\Phi X_t$. □

**Proof of Lemma 3.1.** Take $S = \Omega\left(\frac{L^2 nk \operatorname{Tr}(\Sigma)\ln\left(\frac{T}{\delta}\right)}{\varepsilon^2}\right)$. Letting $M'$ be the matrix such that $y_t - z_t = M'X_t$, and $M = M'\Phi^\top$ as in Theorem 3.3, we have that

$$\mathbb{E}[y_T^{(j)} - z_T] = M'\phi_j = M_j. \tag{9}$$

Let $1 \leq t \leq T$. We bound the error under controls $x_{1:t} \in \mathbb{B}_2^t$, $\|x_{1:t}\|_2 \leq L$ using the triangle inequality. Letting $y_{1:t}$ be the output under $x_{1:t}$,

$$\left\|\mathbb{E}[y_t] - z_t - \hat{M}\Phi X_t\right\|_2 \leq \|\mathbb{E}[y_t - z_t] - M\Phi X_t\|_2 + \|\mathbb{E}[z_t] - z_t\|_2 + \left\|M\Phi X_t - \hat{M}\Phi X_t\right\|_2. \tag{10}$$

By Theorem 3.3, for $k = \Omega(\log^2 T \log(n\rho/\varepsilon))$, choosing constants appropriately, the first term is $\leq \frac{\varepsilon}{4}$.

To bound the second term in (10), we show $z_t$ concentrates around $\mathbb{E}z_t$. We have

$$z_t - \mathbb{E}z_t \sim \eta, \qquad\qquad \eta \sim \mathcal{N}\left(0, \frac{1}{S}\Sigma\right). \tag{11}$$

By concentration of sums of $\chi^2$ random variables (see Hsu et al. (2012), for example),

$$\mathbb{P}_{\eta \sim N(0, \frac{1}{S}\Sigma)}(\|\eta\|_2 \geq \varepsilon') \leq \delta' \qquad \text{as long as } S \geq \frac{5}{\varepsilon'^2}\operatorname{Tr}(\Sigma)\log\left(\frac{3}{\delta'}\right). \tag{12}$$

Take $\delta' = \frac{\delta}{2T}$ and $\varepsilon' = \frac{\varepsilon}{4L\sqrt{nk}}$ and note $S$ was chosen to satisfy (12). Use the union bound to get that

$$\mathbb{P}\left(\exists t \in [1, T], \|z_t - \mathbb{E}z_t\|_2 \geq \frac{\varepsilon}{4L\sqrt{nk}}\right) \leq \frac{\delta}{2}. \tag{13}$$

To bound the third term in (10), we first show that $\hat{M}$ concentrates around $M$. We have

$$\hat{M}_i = \frac{1}{S}\sum_{s=1}^{S}[y_T^{(s,i)}] - z_T \qquad\qquad \forall i \leq nk \tag{14}$$

so

$$\hat{M}_i - M_i = \frac{1}{S}\sum_{s=1}^{S}[y_T^{(s,i)}] - z_T - (\mathbb{E}y_T^{(1,i)} - \mathbb{E}z_T) \tag{15}$$

$$= \eta + (\mathbb{E}z_T - z_T), \qquad\qquad \eta \sim \mathcal{N}\left(0, \frac{1}{S}\Sigma\right) \tag{16}$$

$$\hat{M} - M = \eta' + (\mathbb{E}z_T - z_T)\mathbb{1}^\top, \qquad\qquad \eta' \sim \mathcal{N}\left(0, \frac{1}{S}\Sigma^{\oplus nk}\right). \tag{17}$$

By $\chi^2$ concentration,

$$\mathbb{P}_{\eta \sim \mathcal{N}(0, \frac{1}{S}\Sigma^{\oplus nk})}\left(\|\eta'\|_F \geq \frac{\varepsilon}{4L}\right) \leq \frac{\delta}{2}. \tag{18}$$

We also have $\left\|(\mathbb{E}z_T - z_T)\mathbb{1}^\top\right\|_F \leq \sqrt{nk}\left\|\mathbb{E}z_T - z_T\right\|_2$.

With $\geq 1 - \delta$ probability, we avoid both the bad events in (13) and (18) so $\|z_t - \mathbb{E}z_t\|_2 \leq \frac{\varepsilon}{4L\sqrt{nk}}$ for all $1 \leq t \leq T$ and

$$\left\|\hat{M} - M\right\|_F \leq \frac{\varepsilon}{4L} + \sqrt{nk}\frac{\varepsilon}{4L\sqrt{nk}} = \frac{\varepsilon}{2L} \tag{19}$$

and for all $\|x_{1:t}\| \leq L$, the third term of (10) is bounded by (note $\|\Phi\|_{op} = 1$ because it has orthogonal rows)

$$\left\|M\Phi X_t - \hat{M}\Phi X_t\right\| \leq \left\|M - \hat{M}\right\|_F \|\Phi X_t\|_2 \tag{20}$$

$$\leq \left\|M - \hat{M}\right\|_F \|X_t\|_2 \tag{21}$$

$$\leq \frac{\varepsilon}{2L}L = \frac{\varepsilon}{2}. \tag{22}$$

Thus by (10), $\left\|\mathbb{E}[y_t] - z_t - \hat{M}\Phi X_t\right\|_2 \leq \varepsilon$ with probability $\geq 1 - \delta$. $\square$

## 3.2 Robustness of Control to Uncertainty in Dynamics

To prove Lemma 3.2, we need the following helpful lemma.

**Lemma 3.4.** *For a symmetric LDS $\mathfrak{D}$ with $\rho = \max(\|B\|_F, \|C\|_F, \|D\|_F, \|P\|_{op})$ and $A \preccurlyeq I$, $Q \succcurlyeq \lambda I$, and an approximation $\hat{\mathfrak{D}}^t$ whose predictions $\hat{y}_t$ satisfy the conclusion of Lemma 3.1, we have that for every sequence of controls $(x_1, \ldots, x_T) \in \mathbb{B}_2^T$, and for every $1 \leq t \leq T$,*

$$\left|\sum_{t=1}^T \left[\mathsf{cost}_{\hat{\mathfrak{D}}}^t(x_{1:t}) + \mathrm{Tr}(Q\Sigma) - \mathbb{E}\left[\mathsf{cost}_{\mathfrak{D}}^t(x_{1:t})\right]\right]\right| \leq 2\rho\varepsilon\sqrt{T\min\left(\|\hat{y}_{1:T}\|^2, \|\mathbb{E}y_{1:T}\|^2\right)} + T\rho\varepsilon^2 \tag{23}$$

*where $\mathsf{cost}_{\hat{\mathfrak{D}}}^t(x_{1:t}) = \hat{\mathfrak{D}}^t(x_{1:t})^\top Q\hat{\mathfrak{D}}^t(x_{1:t}) + x_t^\top R x_t$.*

**Proof of Lemma 3.4.** Let $\hat{y}_t = \hat{\mathfrak{D}}_y(x_{1:t})$ and $y_t = \mathfrak{D}_y(\hat{x}_{1:t}) = \mathbb{E}[y_t] + \eta_t$. Using the assumption that $\mathbb{E}[\eta_t\eta_t^\top] = \Sigma$,

$$\left|\mathsf{cost}_{\hat{\mathfrak{D}}}^t(x_{1:t}) + \mathrm{Tr}(Q\Sigma) - \mathbb{E}\left[\mathsf{cost}_{\mathfrak{D}}^t(x_{1:t})\right]\right| \tag{24}$$

$$= \left|\hat{y}_t^\top Q\hat{y}_t + \mathrm{Tr}(Q\Sigma) - \mathbb{E}\left[(\mathbb{E}[y_t] + \eta_t)^\top Q(\mathbb{E}[y_t] + \eta_t)\right]\right| \tag{25}$$

$$= \left|\hat{y}_t^\top Q\hat{y}_t - \mathbb{E}[y_t]^\top Q\mathbb{E}[y_t]\right| \tag{26}$$

$$\leq 2\|\hat{y}_t - \mathbb{E}y_t\|\|Q\|_{op}\min(\|\hat{y}_t\|, \|\mathbb{E}y_t\|) + \|Q\|_{op}\|\hat{y}_t - \mathbb{E}y_t\|^2 \tag{27}$$

$$\left|\sum_{t=1}^T \left[\mathsf{cost}_{\hat{\mathfrak{D}}}^t(x_{1:t}) + \mathrm{Tr}(Q\Sigma) - \mathbb{E}\left[\mathsf{cost}_{\mathfrak{D}}^t(x_{1:t})\right]\right]\right| \tag{28}$$

$$\leq 2\|Q\|_{op}\varepsilon\sqrt{T\min\left(\sum_{t=1}^T \|\hat{y}_t\|^2, \sum_{t=1}^T \|\mathbb{E}y_T\|^2\right)} + T\|Q\|_{op}\varepsilon^2 \tag{29}$$

using Cauchy-Schwarz. $\square$

**Proof of Lemma 3.2.** Define

$$x_{1:T}^* = \operatorname*{arg\,min}_{\substack{x_{1:T} \in \mathbb{B}_2^T \\ \|x_{1:T}\| \leq L}} \mathbb{E}\left[\sum_{t=1}^T \mathfrak{D}_c^t(x_{1:t})\right] \tag{30}$$

$$\hat{x}_{1:T} = \operatorname*{arg\,min}_{\substack{x_{1:T} \in \mathbb{B}_2^T \\ \|x_{1:T}\| \leq L}} \sum_{t=1}^T \hat{\mathfrak{D}}_c^t(x_{1:t}) \tag{31}$$

Let $V = \sum_{t=1}^{T} [\mathbb{E}[\text{cost}_{\mathfrak{D}}^{t}(x_{1:t}^{*})] - \text{Tr}(Q\Sigma)]$, for inputs $x_{1:t}^{*}$. By assumption (3), $V \le cT$. We have

$$V \ge \sum_{t=1}^{T} (\mathbb{E}y_{t})^{T} Q (\mathbb{E}y_{t}) \ge \lambda \|\mathbb{E}y_{1:T}\|^{2} \tag{32}$$

$$\implies \|\mathbb{E}y_{1:T}\| \le \sqrt{\frac{V}{\lambda}}. \tag{33}$$

By Lemma 3.4 for inputs $x_{1:t}^{*}$,

$$\sum_{t=1}^{T} \text{cost}_{\hat{\mathfrak{D}}}^{t}(x_{1:t}^{*}) \le V + 2\rho\varepsilon\sqrt{\frac{TV}{\lambda}} + T\rho\varepsilon^{2} \le V + T\rho\varepsilon\left(2\sqrt{\frac{c}{\lambda}} + \varepsilon\right) \tag{34}$$

Letting $\hat{y}_{1:T}$ be the outputs under $\hat{\mathfrak{D}}$ under the control $\hat{x}_{1:T}$, note that similar to (32),

$$\|\hat{y}_{1:T}\| \le \sqrt{\frac{1}{\lambda}}\sqrt{\mathbb{E}\left[\sum_{t=1}^{T} \text{cost}_{\hat{\mathfrak{D}}}^{t}(\hat{x}_{1:t})\right]} \le \sqrt{\frac{1}{\lambda}}\sqrt{\mathbb{E}\left[\sum_{t=1}^{T} \text{cost}_{\hat{\mathfrak{D}}}^{t}(x_{1:t}^{*})\right]} \tag{35}$$

because $\hat{x}_{1:t}$ is optimal for $\hat{\mathfrak{D}}$. By Lemma 3.4 for inputs $\hat{x}_{1:t}$,

$$\left|\sum_{t=1}^{T} \left[\text{cost}_{\hat{\mathfrak{D}}}^{t}(\hat{x}_{1:t}) + \text{Tr}(Q\Sigma) - \mathbb{E}[\text{cost}_{\mathfrak{D}}^{t}(\hat{x}_{1:t})]\right]\right| \le 2\rho\varepsilon\sqrt{T}\|\hat{y}_{1:T}\| + T\rho\varepsilon^{2}. \tag{36}$$

Now by (34), (35), and (36),

$$\mathbb{E}[\text{cost}_{\mathfrak{D}}(\hat{x}_{1:t})] - \text{cost}_{\mathfrak{D}}(x_{1:t}^{*}) \le (\text{cost}_{\mathfrak{D}}\,\hat{x}_{1:t} + \text{Tr}(Q\Sigma) - \text{cost}_{\hat{\mathfrak{D}}}(\hat{x}_{1:t})) \tag{37}$$

$$+ (\text{cost}_{\hat{\mathfrak{D}}}(\hat{x}_{1:t}) - \text{cost}_{\hat{\mathfrak{D}}}(x_{1:t}^{*})) + (\text{cost}_{\hat{\mathfrak{D}}}(x_{1:t}^{*}) - \text{Tr}(Q\Sigma) - \text{cost}_{\mathfrak{D}}(x_{1:t}^{*})) \tag{38}$$

$$\le 2\rho\varepsilon\sqrt{\frac{T}{\lambda}}\sqrt{V + T\rho\varepsilon\left(2\sqrt{\frac{c}{\lambda}} + \varepsilon\right)} + 0 + \left(T\rho\varepsilon\left(2\sqrt{\frac{c}{\lambda}} + \varepsilon\right)\right) \tag{39}$$

$$= O\left(T\rho\varepsilon\sqrt{\frac{c}{\lambda}}\right) \tag{40}$$

using $\varepsilon \le \frac{\sqrt{c}\lambda}{\rho} \le \sqrt{\frac{c\lambda}{\rho}}$. $\qquad\qquad\qquad\qquad\qquad\qquad\qquad\qquad\qquad\qquad\qquad\qquad\square$

## 4 CONCLUSION

We have presented an algorithm for finding the optimal control inputs for an unknown symmetric linear dynamical system, which requires querying the system only a polylogarithmic number of times in the number of such inputs $T$, while running in polynomial time. Deviating significantly from previous approaches, we circumvent the non-convex optimization problem of system identification by a new learned representation of the system. We see this as a first step towards provable, efficient methods for the traditionally non-convex realm of control and reinforcement learning.

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

## A   AUTOREGRESSIVE MODELS AS A RELAXATION OF LDS

In this section, we verify the statement made in Section 1 on the time and sample complexity of approximating a linear dynamical system with an autoregressive model. Although this is well-known (see, for example, Section 6 of Hardt et al. (2016)), we present a self-contained presentation for convenience and to unify notation.

The *vector autoregressive model with exogenous variables*, or $\mathrm{VARX}(p, s)$, is a touchstone in time-series analysis. Given a time series of inputs (sometimes known as biases) $\{\xi_t\}$, it generates the time series of responses $\{y_t\}$ by the following recurrence:

$$y_t = \sum_{i=1}^{p} A_i y_{t-i} + \sum_{i=0}^{s-1} B_i x_{t-i} + \xi_t.$$

Here, $p$ and $s$ are *memory* parameters, the $\{A_i\}$ and $\{B_i\}$ are matrices of appropriate dimension, and the $\{\xi_t\}$ are noise vectors.

In the special case of $p = 0$, the problem can be solved efficiently with linear regression: in this case, $y_t$ is a linear function of the concatenated inputs $[x_t; x_{t-1}; \ldots; x_{t-s+1}]$.

A $\mathrm{VARX}(0, s)$ model is specified by $M = [M^{(0)}, \ldots, M^{(s-1)}] \in \mathbb{R}^{m \times ns}$ and predicts $y_t = M x_{t:t-s+1}$.

We quantify the relationship between $\mathrm{VARX}(0, s)$ and linear dynamical systems, with a statement analogous to Theorem 3.1:

**Theorem A.1.** *Let $\mathfrak{D}$ be an LDS with size $\mathcal{L}$, fixed $h_1$, and noise $\eta_t = 0$, producing outputs $\{y_1, \ldots, y_T\}$ from inputs $\{x_1, \ldots, x_T\}$. Suppose that the transition matrix of $\mathfrak{D}$ has operator norm at most $\alpha < 1$. Then, for each $\varepsilon > 0$, there is a $\mathrm{VARX}(0, s)$ model with $s = O(\frac{1}{1-\alpha} \log(\mathcal{L}/\varepsilon))$, specified by a matrix $M$, such that*

$$\|y_t - M x_{t:t-s+1}\| \le \varepsilon.$$

*Proof.* By the modification of $\mathfrak{D}$ given in the proof of Theorem 3.3, we may assume without loss of generality that $D = O$. Also, as in the discussion of Theorem 3.1, it can be assumed that the initial hidden state $h_1$ is zero.

Then, we construct the block of $M$ corresponding to lag $i$ as

$$M^{(i)} = CA^i B.$$

This is well-defined for all $1 \le i \le t$. Note that when $s \ge t$, the autoregressive model completely specifies the system $\mathfrak{D}$, which is determined by its (infinite-horizon) impulse response function. Furthermore, by definition of $\alpha$, we have

$$\|M^{(i)}\|_F^2 \le \alpha^i \mathcal{L}^2.$$

Noting that

$$y_t - M x_{t:t-s+1} = \sum_{i=s}^{t-1} M^{(i)} x_{t-i},$$

we conclude that

$$\|y_t - M x_{t:t-s+1}\| \le \sum_{i=s}^{t-1} \alpha^i \mathcal{L}^2 \|x_{t-i}\|_2 \le \frac{\alpha^s \mathcal{L}^2}{1 - \alpha},$$

implying the claim by the stated choice of $s$. $\qquad\square$

$\mathrm{VARX}(0, s)$ only serves as a good approximation of an LDS whose hidden state decays on a time scale shorter than $s$; when the system is ill-conditioned ($\alpha$ is close to 1), this can get arbitrarily large, requiring the full time horizon $s = T$.

On the other hand, it is clear that both the time and sample complexity of learning a $\mathrm{VARX}(0, s)$ model grows linearly in $s$. This verifies the claim in the introduction.

