# OpenReview forum: "Towards Provable Control for Unknown Linear Dynamical Systems"
_ICLR.cc/2018/Conference — Invite to Workshop Track_

### Official Review · AnonReviewer3 · 2017-11-25
**Idea is OK but the paper is not clearly written**

**Rating:** 4
**Confidence:** 3

**Review:**

This paper proposes a new algorithm to generate the optimal control inputs for unknown linear dynamical systems (LDS) with known system dimensions.

The idea is exciting LDS by wave filter inputs and record the output and directly estimate the operator that maps the input to the output instead of estimating the hidden states. After obtaining this operator, this paper substitutes this operator to the optimal control problem and solve the optimal control problem to estimate the optimal control input, and show that the gap between the true optimal cost and the cost from applying estimated optimal control input is small with high probability.
I think estimating the operator from the input to the output is interesting, instead of constructing (A, B, C, D) matrices, but this idea and all the techniques are from Hazan et. el., 2017. After estimating this operator, it is straightforward to use this to generate the estimated optimal control input. So I think the idea is OK, but not a breakthrough.

Also I found the symmetric matrix assumption on A is quite limited. This limitation is from Hazan et. el., 2017, where the authors wants to predict the output. For prediction purposes, this restriction might be OK, but for control purposes, many interesting plants does not satisfy this assumption, even simple RL circuit. I agree with authors that this is an attempt to combine system identification with generating control inputs together, but I am not sure how to remove the restriction on A.
Dean et. el., 2017 also pursued this direction by combining system identification with robust controller synthesis to handle estimation errors in the system matrices (A, B) in the state-feedback case (LQR), and I can see that Dean et. el. could be extended to handle observer-feedback case (LQG) without any restriction.

Despite of this limitation I think the paper's idea is OK and the result is worth to be published but not in the current form. The paper is not clearly written and there are several areas need to be improved.

1. System identification.
Subspace identification (N4SID) won't take exponential time. I recommend the authors to perform either proper literature review or cite one or two papers on the time complexity and their weakness. Also note that subspace identification can estimate (A, B, C, D) matrices which is great for control purposes especially for the infinite horizon LQR.

2. Clarification on the unit ball constraints.
Optimal control inputs are restricted to be inside the unit ball and overall norm is bounded by L. Where is this restriction coming from? The standard LQG setup does not have this restriction.

3. Clarification on the assumption (3).
Where is this assumption coming from? I can see that this makes the analysis go through but is this a reasonable assumption?  Does most of system satisfy this constraint? Is there any? It's ok not to provide the answer if it's hard to analyze, but if that's the case the paper should provide some numerical case studies to show this bound either holds or the gap is negligible in the toy example.

4. Proof of theorem 3.3.
Theorem 3.3 is one of the key results in this paper, yet its proof is just "noted". The setup is slightly different from the original theorem in Hazan et. el., 2017 including the noise model, so I strongly recommend to include the original theorem in the appendix, and include the full proof in the appendix.

5. Proof of lemma 3.1.
I found it's hard to keep track of which one is inside the expectation. I recommend to follow the notation E[variable] the authors been using throughout the paper in the proof instead of dropping these brackets.

6. Minor typos
In theorem 2.4, ||Q||_op is used for defining rho, but in the text ||Q||_F is used. I think ||Q||_op is right.

---

> ### Author Response · Authors · 2017-12-16
> **Response**
>
> 1. We thank the review for pointing this out. However, we did not find clear provable guarantees for N4SID (in terms of sample complexity, etc.) in our setting. If the reviewer were to give a clear reference or explanation, we would be happy to include it.
> Our claim on exponential time is based on the fact that system identification using any kind of local search (ex. gradient descent) converges to a local optimum. It’s not clear how to ensure that the search will reach the actual parameters, beyond a method that takes exponential time such as grid search.
> 3. This condition is now rewritten to be clearer. The assumption $Q>\lambda I$ is reasonable because it says that all directions of the output incur cost - a common case is just $Q=I$. Inequality (3) says that we can incur not much more loss than just the background noise. This is true as long as the system can be driven to 0 in a reasonable amount of time.
> 4. See Main Point 3.
> 5. Done.
> 6. Done.

---

### Official Review · AnonReviewer1 · 2017-11-27
**Review of "Towards Provable Control for Unknown Linear Dynamical Systems"**

**Rating:** 7
**Confidence:** 3

**Review:**

This paper studies the control of symmetric linear dynamical systems with unknown dynamics. Typically this problem is split into a (non-convex) system ID step followed by a derivation of an optimal controller, but there are few guarantees about this combined process.  This manuscript formulates a convex program of optimal control without the separate system ID step, resulting in provably optimality guarantees and efficient algorithms (in terms of the sample complexity).  The paper is generally pretty well written.

This paper leans heavily on Hazan 2017 paper (https://arxiv.org/pdf/1711.00946.pdf). Where the Hazan paper concerns itself with the system id portion of the control problem, this paper seems to be the controls extension of that same approach.   From what I can tell, Hazan's paper introduces the idea of wave filtering (convolution of the input with eigenvectors of the Hankel matrix); the filtered output is then passed through another matrix that is being learned online (M). That matrix is then mapped back to system id (A,B,C,D).  The most novel contribution of this ICLR paper seems to be equation (4), where the authors set up an optimization problem to solve for optimal inputs; much of that optimization set-up relies on Hazan's work, though. However, the authors do prove their work, which increases the novelty. The novelty would be improved with clearer differentiation from the Hazan 2017 paper.

My biggest concerns that dampen my enthusiasm are some assumptions that may not be realistic in most controls settings:

- First, the most concerning assumption is that of a symmetric LDS matrix A (and Lyapunov stability).  As far as I know, symmetric LDS models are not common in the controls community.  From a couple of quick searches it seems like there are a few physics / chemistry applications where a symmetric A makes sense, but the authors don't do a good enough job setting up the context here to make the results compelling. Without that context it's hard to tell how broadly useful these results are.  In Hazan's paper they mention that the system id portion, at least, seems to work with non-symmetric, and even non-linear dynamical systems (bottom of page 3, Hazan 2017). Is there any way to extend the current results to non-symmetric systems?

- Second, it appears that the proposed methods may rely on running the dynamical system several times before attempting to control it.  Am I misunderstanding something?  If so this seems like it may be a significant constraint that would shrink the application space and impact even further.

---

> ### Author Response · Authors · 2017-12-16
> **Response**
>
> Re: innovation compared to HSZ’17: The reviewer asked whether LDS control is a simple consequence of the ability to predict the next reward, as shown in HSZ17. This issue confused us too originally. But prediction in the sense of HSZ17 is a lot easier because the guarantee is in terms of mean-squared error for a single input-output sequence, over a large number of steps.  Such MSE error permits predictions to be off for long stretches of time. To do control on the other hand one needs to look ahead at results of all control choices up to the horizon L and pick the best. Since the HSZ17 predictions for different lookahead paths may have arbitrary error in any time interval, the estimate for the max reward over all paths can be arbitrarily off. The bulk of the paper is showing that it is nevertheless possible with small sample complexity, and the proof is novel over HSZ17.
>
> 1. The assumption that the LDS uses a *symmetric* matrix is indeed crucial for our result. However, note that solving the symmetric case is still significant progress on the problem of provably efficient control of LDS, which has been open for decades.
>
> 2. The reviewer is correct that our proposed methods will rely on running the dynamical system several times. The need for multiple restarts is inherent to the problem of learning the system, at least under the assumptions in our setting. Notice that one cannot simply wait for the state to decay, since the transition matrix can have an eigenvalue of 1. A basic example shows this: suppose A is a tridiagonal matrix, B controls the first dimension of h, C observes the last dimension of h. Then, multiple restarts are needed to find the optimal control, since there is a delay before C can be determined. We will update the appendix with the full construction, to clarify this point. See also Main Point 1.

---

### Official Review · AnonReviewer2 · 2017-12-08
**Interesting approach but maybe more suited for a theory conference (no experiments).**

**Rating:** 5
**Confidence:** 4

**Review:**

The paper presents a provable algorithm for controlling an unknown linear dynamical system (LDS). Given the recent interest in (deep) reinforcement learning (combined with the lack of theoretical guarantees in this space), this is a very timely problem to study. The authors provide a rigorous end-to-end analysis for the LDS setting, which is a mathematically clean yet highly non-trivial setup that has a long history in the controls field.

The proposed approach leverages recent work that gives a novel parametrization of control problems in the LDS setting. After estimating the values of this parametrization, the authors formulate the problem of finding optimal control inputs as a large convex problem. The time and sample complexities of this approach are polynomial in all relevant parameters. The authors also highlight that their sample complexity depends only logarithmically on the time horizon T. The paper focuses on the theoretical results and does not present experiments (the polynomials are also not elaborated further).

Overall, I think it is important to study control problems from a statistical perspective, and the LDS setting is a very natural target. Moreover, I find the proposed algorithmic approach interesting. However, I am not sure if the paper is a good fit for ICLR since it is purely theoretical in nature and has no experiments. I also have the following questions regarding the theoretical contributions:

(A) The authors emphasize the logarithmic dependence on T. However, the bounds also depend polynomially on L, and as far as I can tell, L can be polynomial in T for certain systems if we want to achieve a good overall cost. It would be helpful if the authors could comment on the dependence between T and L.

(B) Why does the bound in Theorem 2.4 become worse when there are some directions that do not contribute to the cost (the lambda dependence)?

(C) Do the authors expect that it will be straightforward to remove the assumption that A is symmetric, or is this an inherent limitation of the approach?

Moreover, I have the following comments:

(1) Theorem 3.3 is currently not self-contained. It would enhance readability of the paper if the results were more self-contained. (It is obviously good to cite results from prior work, but then it would be more clear if the results are invoked as is without modifications.)

(2) In Theorem 1.1, the notation is slightly unclear because B^T is only defined later.

(3) In Section 1.2 (Tracking a known system): "given" instead of "give"

(4) In Section 1.2 (Optimal control): "symmetric" instead of "symmetrics"

(5) In Section 1.2 (Optimal control): the paper says "rather than solving a recursive system of equations, we provide a formulation of control as a one-shot convex program". Is this meant as a contrast to the work of Dean et al. (2017)? Their abstract also claims to utilize a convex programming formulation.

(6) Below Definition 2.3: What is capital X?

(7) In Definition 2.3: What does the parenthesis in \phi_j(1) denote?

(8) Below Theorem 2.4: Why is Phi now nk x T instead of nk x nT as in Definition 2.3?

(9) Lemma 3.2: Is \hat{D} defined in the paper? I assume that it involves \hat{M}, but it would be good to formally define this notation.

---

> ### Author Response · Authors · 2017-12-16
> **Response**
>
> A. For reasonable systems L is a constant. See Main Point 3.
> B. If there are directions that do not contribute to the cost, then under the optimal control, the output may be large in that direction. Our bounds for the error depend on the size of the outputs y (Lemma 3.4) because the error in estimating the quadratic form depends linearly on the size of y.
> C. This requires further work. We have ongoing work on extending the work of Hazan, Singh, and Zhang to the nonsymmetric case, which will then also allow control.
> 2. Fixed.
> 3. Fixed.
> 4. Fixed.
> 5. See Main Point 2.
> 6. Should be x. Fixed.
> 7. phi_j(k) denotes the kth entry of \phi_j.
> 8. Typo, fixed.
> 9. \hat{D} is exactly the analogue of D for the predicted dynamics.

---

### Author Response · Authors · 2017-12-16
**Main points**

We thank the reviewers for their comments, and note the following main points.

1. The difference between this paper and [HSZ17] is as follows. The results of [HSZ17] together with random exploration requires sample complexity that scales with poly(T). We show how to explore better with the filters than with random exploration, significantly reducing sample complexity to polylog(T), This is an important point, since poly(T) bounds can be obtained by straightforward regression and can be considered folklore.

2. Our work is distinguished from Dean et al’s work as follows:
The Dean et al. work considers a case with no hidden state - this is known to be efficiently solvable by convex optimization.
In contrast, our setting is more general and has an evolving hidden state. The natural formulation is thus via *non-convex* optimization, for which no efficient algorithm was known before to our work.

3. Clarification on the unit ball constraints (Optimal control inputs are restricted to be inside the unit ball and overall norm is bounded by L):
The constraint comes from the fact that the error from the learned dynamics scales as the input.
Unit ball constraint: This is a reasonable setting because often there is a maximum input that one can put into the system. It is without loss of generality because for the unrestricted setting, for a reasonable system starting at a bounded hidden state, the optimal control input will be bounded by some norm, which can be rescaled to 1. (Just scale down by an upper bound on the norm.)
Overall norm constraint: This is reasonable because when the system is controllable, the optimal control decays the state geometrically, and the total sum of inputs is bounded.

---

### Decision · Program_Chairs · 2018-01-29
**ICLR 2018 Conference Acceptance Decision**

**Decision:**

Invite to Workshop Track

**Comment:**

This paper studies the control of symmetric linear dynamical systems with unknown dynamics. While the reviewers agree that this is an interesting topic, there are concerns that the assumptions are not realistic. Lack of experiments also stands out. I recommend the paper to workshop track with the hope that it will foster more discussions and lead to more realistic assumptions.